# Interpreting Reward Models in RLHF-Tuned Language Models Using Sparse Autoencoders

## Abstract

Large language models (LLMs) aligned to human preferences via reinforcement learning from human feedback (RLHF) underpin many commercial applications of LLM technology. Despite this, the impacts of RLHF on LLM internals remain opaque. We propose a novel method for interpreting implicit reward models (IRMs) in LLMs learned through RLHF. Our approach trains pairs of autoencoders on activations from a base LLM and its RLHF-tuned variant. Through a comparison of autoencoder hidden spaces, we identify features that reflect the accuracy of the learned reward model. To illustrate our method, we fine-tune an LLM via RLHF to learn a token-utility mapping and maximize the aggregate utility of generated text. This is the first application of sparse autoencoders to interpreting IRMs. Our method provides an abstract approximation of reward integrity and holds promise for measuring alignment between specified objectives and learned model behaviors.

## 1 Introduction

Do implicit reward models (IRMs) learned by Large Language Models (LLMs) through Reinforcement Learning from Human Feedback (RLHF) diverge from their intended training objectives? How can we interpret these IRMs and measure such divergences?

LLMs are commonly fine-tuned with RLHF to align outputs with a reward measure. Despite the widespread adoption of RLHF, it remains opaque how well the student model internalizes the explicit reward function, making failures in the IRM difficult to detect. Contributing to this difficulty is superposition in the features used in LLMs (Elhage et al., 2022a), as well as full model interpretability research being at an early stage.

As LLMs steered via RLHF scale in capability and deployment, the implications of failures in the IRM amplify. Misspecified rewards can cause 'specification gaming' (Krakovna et al. (2020)), whereby a model engages in an undesired behavior while still achieving high reward. Through behaviors like sycophancy, this phenomenon can be observed to be emerging in LLMs already (Wei et al. (2023)). Other risks include manipulation of the user's preferences (Adomavicius et al. (2013)), reinforcement of the biases present in human labellers (Santurkar et al. (2023)) and potentially catastrophic outcomes in situations where models approach or generally exceed human capabilities (Christiano (2019)). Detecting such failures of RLHF in the wild is challenging, as models may be incentivized to appear more aligned than they are (Hubinger et al. (2019)) in an effort to preserve their reward model(s) (Omohundro (2008)).

In this work, we present a novel technique to interpret IRMs learned through RLHF. While prior work has applied sparse coding to derive more interpretable features from LLMs (Sharkey et al. (2022); Cunningham et al. (2023)), we extend those methods to IRMs, proposing their use for IRM interpretation and measurement. Our major contribution is applying sparse coding towards (a) distinguishing features that specifically emerge from the RLHF tuning process and (b) quantifying the accuracy of the learned IRM in matching the preferences of the overseer during fine-tuning. To the best of our knowledge, our paper is the first to apply sparse coding to the study of reward models.

Our procedure can be broken down into the following steps, also illustrated in Figure 1.

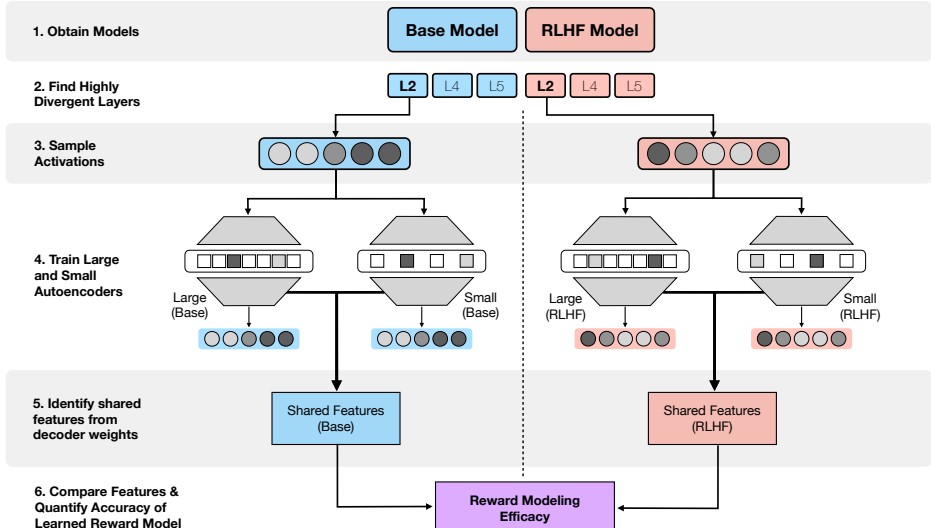

Figure 1: First, we sample activations from layers having the highest parameter divergence between $M_{\text{base}}$ and $M_{\text{RLHF}}$. Then, two autoencoders with a sparsity constraint are trained on those activations, each with a different dictionary size. The overlap is computed between the two dictionaries to find high-confidence features that serve as a proxy for ground truth. We analyze activations on these features, enabling both manual inspection of features as well as computing an aggregate score for the implicit reward model.

1. **Find Highly Divergent Layers:** After RLHF, compute the parameter divergence between the base model $M_{\text{base}}$ and the fine-tuned model $M_{\text{RLHF}}$, and sort layers in descending order by divergence. Given that if an IRM is learned it must be encoded by the differences in parameters between $M_{\text{base}}$ and $M_{\text{RLHF}}$, we avoid training useless autoencoders (for the task of IRM interpretation) by discarding layers unlikely to contain components of the IRM.

2. **Train Large and Small Autoencoders:** Train an autoencoder with a sparsity constraint on activations from $M_{\text{RLHF}}$ over a an unseen corpus for the top-$n$ layers with the highest parameter divergence to construct a hidden space feature representation, and then another autoencoder with a smaller dictionary size. Do the same for the corresponding layers in $M_{\text{base}}$.

3. **Identify Shared Features:** Compute overlapping features across the larger and smaller learned dictionaries for both autoencoder pairs, to identify ground truth features in $M_{\text{base}}$ and $M_{\text{RLHF}}$.

4. **Compare Features and Quantify IRM Efficacy:** Compare the differences in features identified in $M_{\text{base}}$ and $M_{\text{RLHF}}$, such that an interpretable notion of the effects of RLHF on $M_{\text{base}}$ is attained through the relative feature differences. We later use these features in a quantitative measure of the efficacy of the internal reward model, as well as in qualitative analysis.

## 2 BACKGROUND

**Mechanistic Interpretability**   Understanding the inner workings of neural networks such as transformers is essential for fostering transparency and trust. In recent years, mathematical frameworks have been developed to represent and analyze the computations within these models (Elhage et al., 2022b). Foote et al. (2023); Bills et al. (2023) offer another approach whereby a larger model predicts what human-interpretable concept a neuron might

represent in a smaller model. For a different perspective, Black et al. (2022) construct the 'polytope lens', which proposes polytopes as the fundamental interpretable units of a neural network instead of individual neurons or linear combinations of them. These frameworks propose scalable methods for describing the internal functioning of transformers, enabling transparency in the model's functioning and the verification of properties useful for safety, like accurate reward modeling.

Our work interprets the internals of transformer-based LLMs with a vocabulary size $V$. The models take an input sequence $(\mathtt{x}_1, \ldots, \mathtt{x}_p)$ where each $\mathtt{x}_i \in \{1, \ldots, V\}$. Tokens are mapped to $\mathtt{d}_e$-dimensional embeddings by selecting the $\mathtt{x}_i$-th column of an embeddings matrix $\mathrm{Embd} \in \mathbb{R}^{d_e \times V}$.

**External Reward Models in RLHF**   RLHF has emerged as the dominant paradigm for fine-tuning large language models to represent human preferences. It is performant even if the desired behavior is complex or not easily quantifiable, making it significantly more effective than hand-crafted reward functions.

In common RLHF settings, a dataset of human comparisons between outputs of the base model is first collected, providing feedback on which outputs are preferable (Christiano et al. (2023); Ziegler et al. (2020)). In the Reinforcement Learning through AI Feedback (RLAIF) variation of the fine-tuning scheme, this dataset is AI-generated, removing the need for human participation in the fine-tuning process (Bai et al. (2022)).

This dataset is used to train a reward model to predict human preference scores, replacing traditional reward functions. In the context of language models, this reward model is often itself a separate instance of an LLM. The reward model is used to fine-tune the policy of the base model. Techniques like proximal policy optimization (Schulman et al. (2017)) are commonly employed to optimize the policy model using scores under the reward model as the objective. By the end of a successful fine-tuning process, the policy model has internalized an implicit model of the external preferences (human feedback, in the case of RLHF).

In this paper, we analyze the implicit reward model (IRM) internalized by the policy model. To differentiate clearly, we will refer to the reward models used to oversee the RLHF process as external reward models (ERMs).

**Feature Superposition in Deep Learning Models**   There is a significant body of evidence indicating that deep neural networks learn human-interpretable features of the input (Bills et al., 2023; Karpathy et al., 2015; Olah et al., 2017; Mikolov et al., 2013). By features, we mean vectors in a network's activation space that correspond to human-understandable concepts, such as apostrophes or arithmetic. Often, deep neural networks store the features in a distributed way; as a result, individual neurons do not correspond to a single semantic feature. This phenomenon has been coined "superposition" (Elhage et al. (2022a)). It allows a model to represent more features than it has dimensions in its activation space, especially when those features are sparsely present in training data. Superposition poses a major obstacle to neural network interpretability, and this is expected to extend to the interpretation of reward models learned through RLHF in LLMs.

**Sparse Autoencoders for Activation Vector Reconstruction**   Autoencoders minimize the reconstruction error $\epsilon$ for an input vector $x$ subject to projection into a latent space:

$$\epsilon = \|x - \mathrm{Dec}(\mathrm{Enc}(x))\|^2 \qquad (1)$$

Enc represents the encoding function, and Dec the decoding function. For activation vectors, sparse autoencoders constrain the activations in the hidden layer $h$ to a limited number $k$ of active neurons, and we stipulate the encoding function Enc to be $\mathrm{Enc}_k$ in this case.

As a result of the sparsity constraint on the autoencoder, each vector in Dec encodes a handful of neurons from the activation vector. A compressed representation capturing key activation patterns emerges, identifying 'ground truth features' in the model that activations were sampled from. Early results from Sharkey et al. (2022) and Cunningham et al. (2023)

suggest sparse autoencoders can recover ground truth features, even when those features are represented in a superposed manner.

**Autoencoder Architecture**   Our autoencoder architecture consists of an encoder, composed of a linear layer preceding a ReLU activation function, and a linear decoder. Sparsity in the decoder is induced through $L_1$ regularization on the weights, forcing the network to learn a sparser representation.

The decoder and encoder weights are tied. Prior to being encoded, the weights are normalized to have unit norm. The overall loss function is calculated as the sum of the mean squared error between the reconstructed output from the decoder and the true data (for both training the decoder and measuring performance) and an $L_1$ loss term on the decoder weight matrix. We scale the $L_1$ loss by an $L_1$ coefficient, to tune the importance given to sparsity. This architecture is based on the experimental results of Sharkey et al. (2022).

**Deducing Features From Dictionary Similarities Between Autoencoders of Different Sizes**   Sharkey et al. (2022) identify features in toy models exhibiting superposition by training two sparse autoencoders of different sizes, and taking a similarity measurement between the decoder weights of the two autoencoders. They show that features with high similarity between the two learned dictionaries (the decoder weights matrix) correspond to ground truth features exhibited in the transformer. These results are corroborated by Cunningham et al. (2023) where the same technique is applied to language models, showing best-in-class performance.

For their similarity measure between two learned dictionaries, Sharkey et al. (2022) define 'Mean Max Cosine Similarity' (MMCS). Let $D$ and $D'$ be two dictionaries, and $d$ and $d'$ be elements from each dictionary. Then we have:

$$\text{MMCS}(D, D') = \frac{1}{|D|} \sum_{d \in D} \max_{d' \in D'} \text{CosineSim}(d, d'). \tag{2}$$

Intuitively, MMCS is just the average nearest neighbor similarity for features to $D$ from $D'$. In the above, let $D_g$ be the top $k$ features of $D$ that realize the highest contribution to the MMCS. In the case of LLMs, the ground truth features are unknown, and so the set $D_g$ is used as a proxy for a true representation of the ground truth features.

**Automating Neuron Interpretability Using Large Language Models**   Identifying plausible descriptions of what a given neuron represents is laborious for a human. Thus, approaches like Bills et al. (2023); Foote et al. (2023) automate this process. Bills et al. (2023) provide GPT-4 with a set of normalized (to a range of 0 and 10, where 10 indicates maximal activation) and discretized activations for a set of tokens passed to the model as a prompt. GPT-4 then predicts an explanation for what the neuron represents based on those activations, and then simulates discretized activations for tokens as if that description were true.

## 3   RELATED WORK

To our knowledge, no general methods have been proposed for finding human-interpretable representations of IRMs learned via RLHF and RLAIF. Nevertheless, there have been works in similar domains.

Jenner & Gleave (2021) provide a framework for preprocessing reward functions learned by RL agents into simpler but equivalent reward functions, which makes visualizations of these functions more human-understandable. Michaud et al. (2020) explain the reward functions learned by Gridworld and Atari agents using saliency maps and counterfactual examples, and find that learned reward functions tend to implement surprising algorithms relying on contingent aspects of the environment. They also note that reward interpretability requires a different set of tools from policy interpretability. We share with these works the desire to

find new general tools for reward model interpretability, but focus on reward models learned through RLHF and RLAIF rather than standard RL training.

Furthermore, Gleave et al. (2021) and Wolf et al. (2023) present methods for comparing and evaluating learned reward functions in the standard RL setting without requiring these functions to be human-interpretable. In comparison, we aim for evaluation of IRMs in the RLHF setting through interpretability.

There is also existing literature on circumventing superposition when interpreting deep learning models. Olah et al. (2020) introduce the problem of superposition and its effect on interpretability. Elhage et al. (2022a) present a toy model where the superposed features can be fully understood and outline possible directions for tackling the problem in real-world models. One of the proposed approaches, sparse dictionary learning (Olshausen & Field (1997), Lee et al. (2006)) to find directions in the activation space that correspond to features, also forms the basis of our work.

Sharkey et al. (2022) present a report of preliminary attempts to apply sparse dictionary learning on deep neural networks. Cunningham et al. (2023) build upon the work of Sharkey et al. (2022), finding that the dictionary features learned by sparse autoencoders are more amenable to automated interpretability techniques introduced by Foote et al. (2023); Bills et al. (2023). They also find that the dictionary features are more precise and monosemantic compared to features brought out of superposition by other methods, such as principal component analysis (Wold et al. (1987)) and independent component analysis (Lee (1998)). Their experiments are conducted on Pythia-70M language models, but in comparison to our work, do not assess whether this method is applicable to learned reward models.

Other works exploring related techniques include Yun et al. (2021), who apply sparse dictionary learning to visualize the residual streams of transformer models, and Gurnee et al. (2023), who find human-interpretable features in large language models using sparse linear probes. Finally, an alternative approach for circumventing superposition has been explored by Jermyn et al. (2022), who engineer models to have more monosemantic neurons by intervening in the training process and changing the local minimum the model's weights converge to.

## 4 METHODOLOGY

### 4.1 INTERPRETING LEARNED REWARD MODELS IN LLMS.

Our primary method for interpreting IRMs learned through RLHF and RLAIF consists of first isolating LLM layers relevant to reward modeling, then using sparse autoencoders to reconstruct activation vectors from these layers, and finally using GPT-4 to reconstruct feature explanations for the activation vectors. This can be separated into the following components:

- Identify the set of layers $L$ in an RLHF-tuned LLM $M_{\text{RLHF}}$ likely related to the learned IRM. We do so by sorting layers in order of increasing magnitude of $\Delta(L_{M_{\text{RLHF}}}, L_{M_{\text{base}}})$, where $\Delta$ is the sum of Euclidean distances between each corresponding weight and bias tensor in the layer between $M_{\text{RLHF}}$ and the corresponding base model $M_{\text{base}}$. In the following bullets, we simplify notation by describing our feature extraction for a single fixed layer $\ell$ of $L$.

- For both $M_{\text{RLHF}}$ and $M_{\text{base}}$, train two autoencoders, $\mathcal{AE}_1$ and $\mathcal{AE}_2$, of differing hidden sizes, and with the same sparsity constraint. These autoencoders reconstruct activation vectors (obtained through prompting with the test split of the relevant dataset) on $\ell$ for their respective model (Sharkey et al. (2022); Cunningham et al. (2023)). For *each* model, we extract a pair of lower-dimensional feature dictionaries, $D_1$ and $D_2$, from the corresponding autoencoder. Each feature is a column of the decoder's weight matrix.

- Because autoencoders produce varying dictionaries over training runs and hyperparameters, we keep only the features that occur in both $D_1$ and $D_2$. We compute the

MMCS between $D_1$ and $D_2$ in order to identify repeating features across the two dictionaries, indicating that shared features truly occur in the model.

- The top-$k$ most similar features between $D_1$ and $D_2$ in terms of MMCS are explained using a variation of the method by Bills et al. (2023) designed to directly describe the features in a dictionary. The method feeds the encoder of $\mathcal{AE}_n$ activations from the model on which it was trained, and then GPT-4 predicts a description of that feature from the feature weights specified in the encoder output.

- By comparing these explanations in $M_{\text{RLHF}}$ and $M_{\text{base}}$, we show how these descriptions can be correlated with the efficacy of the IRM in encapsulating the explicit reward model.

- This method is applied to a training regime in which $M_{\text{RLHF}}$ is tasked with learning an explicit table of words and maximizing their presence within PPO training. This training environment allows us to quantitatively assess the efficacy of $M_{\text{RLHF}}$'s reward model.

### 4.2 Overseer-Guided Fine-Tuning Using Utility Tables.

As a case study, we construct a fine-tuning environment simpler than conventional RLHF. An overseer, denoted as $O$, is imbued with a "utility table" $U$: a mapping of words to respective utility values. The overseer converts a tokenized generation to words, and then computes the utility of the generation and prefix together.

The aim is to modulate the student model, $M_{\text{RLHF}}$, to maximize the utility of its output text. Utility values are assigned to tokens in $M_{\text{RLHF}}$'s vocabulary, and we use Proximal Policy Optimization (PPO) for reward training. See Appendix C for more details on the general PPO method, and see Appendix D for more details on the Utility tables task.

We flesh out further details of our setup in Section 5 and lightly explore alternate options in Appendix H.

### 5 Experiments

We detail here each stage of our experimental pipeline, from training LLMs via RLHF, to extracting dictionary features from autoencoders, to finally interpreting the IRMs using these dictionary features.

### 5.1 Applying RLHF to base models.

We select a controlled sentiment generation task using data from the IMDb reviews dataset due to the simplicity of the training environment, reducing noise in our analysis. Models generate completions to review prefixes, and positive sentiment prefix and completion pairs are assigned higher rewards. Two different external sentiment reward models are used for fine-tuning via RLHF.

The first is a DistilBERT (Sanh et al., 2020) sentiment classifier trained on the IMDb reviews dataset (von Werra, 2023). Reward is assigned to the logit of the positive sentiment label. The second is the Utility table reward model described in Section 4.2, where the utility values are taken from the VADER sentiment lexicon (Hutto & Gilbert (2014)). The sentiment values were initially labelled by a group of human annotators, who assigned ratings from $-4$ (extremely negative) to $+4$ (positive), with an average taken over ten annotations per word. We assigned reward to a sentence as a sum of utilities, scaled down by a factor of 5 and clamped to an interval of $[-10, 10]$. Scaling and clamping were implemented to avoid collapse in PPO training, which was observed if reward magnitudes were left unbounded.

$$\text{Reward}(s) = \text{clip}\left(\frac{1}{5} \sum_{\text{token} \in s} U(\text{token}), -10, +10\right) \tag{3}$$

Our experiments are run with various models from the Pythia suite (70M, 160M and 410M) (Biderman et al., 2023). These models are fine-tuned with equivalent hyperparameters via Proximal Policy Optimization (PPO), in a setup akin to Ouyang et al. (2022). For fine-tuning, we used the Transformers Reinforcement Learning (TRL) framework (von Werra et al., 2023). The major hyperparameters are listed in Table 1, with the rest derived from the default values provided by the TRL framework. See Appendix C for an overview of the RLHF training pipeline.

Table 1: Hyperparameters used to train models for positive sentiment completions of prefixes from the IMDb dataset.

| Batch Size | Mini Batch Size | Init KL Coef | Max Grad Norm | Learning Rate |
|---|---|---|---|---|
| 64 | 16 | 0.5 | 1 | $1 \times 10^{-6}$ |

### 5.2 Training autoencoders for dictionary extraction

Once we obtain the trained policy model, we compute the parameter divergence between $M_{\mathrm{RLHF}}$ and $M_{\mathrm{base}}$ layer by layer under the $\ell_2$ norm. We sort all layers in descending order from most to least parameter divergence, and fix the five highest-divergence layers for our dictionary extraction. These turned out to mostly be the deeper layers of the models; see Appendix G for details.

For each model from $M_{\mathrm{base}}$ and $M_{\mathrm{RLHF}}$, we train a pair of autoencoders on the activations of each high-divergence layer using two different dictionary sizes. The dictionary sizes in Table 2 were used for the autoencoders.

Autoencoders were trained for 3 epochs with an $L_1$ regularization coefficient of 0.001, a learning rate of $1e-3$ and a batch size of 32 on activations from inputs for the test split of the IMDb reviews dataset. We found that for GPT-Neo-125, an $L_1$ regularization coefficient of 0.0015 gave a better tradeoff of reconstruction and sparsity. The decoder and encoder weights were tied, and the decoder weights are simply a transpose of those for the encoder. 'These hyperparameters were chosen based on empirical testing by Sharkey et al. (2022), Cunningham et al. (2023) and ourselves in selecting for optimal sparsity and reconstruction loss, where we optimized for both the $\ell_1$ and $\ell_0$ sparsity of the dictionary elements.

Next, we find and retain the top $k = 10$ features that maximize the MMCS objective (given earlier in Equation 2) between $D_1$ and $D_2$ of each such pair of autoencoders.

For more discussion on the methodology used to train the autoencoders, see Appendix H.

### 5.3 Measuring fidelity of sparse coding features to the specified reward function.

In order to derive human-interpretable explanations, we employ GPT-4 to explain what a dictionary feature represents based on normalized and discretized activations for that feature (Bills et al. (2023)) over a series of tokens. The top $k = 10$ highest MMCS features were sampled for both $M_{\mathrm{base}}$ and $M_{\mathrm{RLHF}}$ to locate feature indices to explain with GPT-4. Through these explanations of likely ground truth dictionary features, we attempt to understand

Table 2: Dictionary sizes for autoencoder comparison via MMCS

| Model | Dictionary Size | |
|---|---|---|
| | Large | Small |
| Pythia-70m | 1024 | 512 |
| Pythia-160m | 1536 | 768 |
| Pythia-410m | 2048 | 1024 |
| GPT-Neo-125m | 768 | 1536 |

the effects of RLHF on $M_{\text{base}}$, using examples to substantiate analysis in Section 6. See Appendix E for a complete list of feature descriptions for layer 2 in Pythia-70m.

**Reconstructing external Utility table reward model**  Through the sparse coding feature extraction from $M_{\text{RLHF}}$ and subsequent GPT-4 interpretation, we would expect to rederive tokens present in our originally specified utility table $U$ if RLHF is successful influencing the model to learn $U$. For example, if $U$ specifies positive utility tokens (e.g., 'good', 'happy', etc.) and these tokens are more prevalent in the feature descriptions for $M_{\text{RLHF}}$ than in $M_{\text{base}}$, it would indicate $M_{\text{RLHF}}$ having learned this skew.

To quantify the correspondence of the dictionary features to the specified external reward model, we also measure the summed absolute utility of the top-$k$ most similar feature descriptions for both the $M_{\text{base}}$ and $M_{\text{RLHF}}$ dictionaries. We can then use GPT-4's description of these features and the summed absolute utility of these text descriptions to answer: How well has $M_{\text{RLHF}}$ learned $U$?

## 6 RESULTS AND DISCUSSION

In this section, we present a qualitative analysis of the feature explanations generated via GPT-4 for the implicit reward models under both of our tasks. We also give a quantitative measure of the utility of the dictionary features for our Utility table reward model.

### 6.1 MOVIE OPINION FEATURES IN PYTHIA-70M FINE-TUNED ON POSITIVE MOVIE REVIEW COMPLETIONS

Features identified as detecting opinions concerning movies in itself serves as a great example of both the utility and shortcomings of this method. Being able to detect the occurrence of an opinion regarding a movie provides useful insights about the reward model, given that the training objective was generating positive sentiment completions. However, the descriptions of such features are very high-level and overrepresented among the feature descriptions. In the fine-tuned Pythia-70m instance, of the 50 highest similarity features (10 per layer), there are 21 feature descriptions that mention detecting opinions or reviews in the context of movies. Of the top-$k = 10$ features in layer 4 of the fine-tuned model, 8 are for this purpose. Contrast this to the base model, with 13 total feature descriptions focused on sentiment in the context of movie reviews. Full feature description tables are available in Appendix E.

This data alone does not allow for a clear picture of the reward model to be constructed. Although it is clear that a greater portion of the features represent concepts related to the training objective in this limited sample, it cannot be shown that the model has properly internalized the reward model on which it was trained. Additionally, it is highly improbable for the base model to inherently have 13 of the 50 sampled features applied to identifying opinions on movies, which shows that the nature of the input data used to sample activations can skew GPT-4's description of the feature. If a feature consistently activates on negative opinions, but the entire sample set is movie reviews, it might be unclear to GPT-4 whether the feature is activating in response to negative sentiment, or to negative sentiment in movie reviews specifically. This underscores the need for future work to use a diverse sample of inputs when sampling activations for use in this method. The next case study tries to cover a quantitative metric for reward modeling efficacy, but also falls short of showing a crisp structure of elements comprising the reward model.

### 6.2 QUANTIFYING REWARD MODELING EFFICACY FOR MODELS FINE-TUNED ON HIGH UTILITY MOVIE REVIEW COMPLETIONS

Not all dictionary features will be relevant to the utility table. Using the example of 'Sentences concerning word processing' as a feature description, it is not obvious how the utility of this could be computed under any $U$. Sentiment lexicons like VADER lend themselves well to this task. Neutral entries are labeled as having a sentiment score of 0, and words not included in the lexicon are treated as though they were neutral entries. A quantitative measure is attempted in Table 3, whereby GPT-4's predicted explanations are computed against $U$ for

an approximation of $M_{\mathrm{RLHF}}$'s ability to learn $U$ and its maximization. This metric is shown alongside the aggregate utility measured over 100 completions of a 30 token prefix sampled from the test split to validate it as correlating with actual performance against the reward model. See Table 4.

Table 3: Mean of the aggregate absolute utility of the top-$k = 30$ learned features in the base and fine-tuned model over three samples per model

| Model | $M_{\mathbf{base}}$ **Score** | $M_{\mathbf{RLHF}}$ **Score** |
| --- | --- | --- |
| Pythia-70m | 61.2 | 94.3 |
| Pythia-160m | 59.2 | 80.2 |
| Pythia-410m | 59.4 | 89.4 |
| GPT-Neo-125m | 101.2 | 111.0 |

Table 4: Aggregate absolute utility of 100 completions to 30 token prefixes for the base and fine-tuned models

| Model | $M_{\mathbf{base}}$ | $M_{\mathbf{RLHF}}$ |
| --- | --- | --- |
| Pythia-70m | 68.8 | 137.6 |
| Pythia-160m | 103.1 | 172.1 |
| Pythia-410m | 108.5 | 212.0 |
| GPT-Neo-125m | 104.3 | 115.1 |

The descriptions of the top-$k$ represented features score considerably more highly in $U$, suggesting a superior IRM. Although *indicative* of what features might compose the reward model of $M_{\mathrm{RLHF}}$, the accuracy of this method is limited by two primary factors: the capability of the sparse autoencoders in reconstructing accurate activation vectors, and GPT-4's ability to accurately devise descriptions for neurons. Additionally, aggregating the absolute utility of feature descriptions is simply a proxy for reward modeling efficacy, and is not guaranteed to map to equivalent performance against $U$ empirically.

## 7 CONCLUSION

In closing, features contained in the dictionaries of autoencoders specific to our fine-tuned model, $M_{\mathrm{RLHF}}$, are explained using GPT-4. Explanations that imply properties of the reward model are used as case studies to demonstrate their usefulness for studying the reward models learned through RLHF. Additionally, we quantify the efficacy of the reward model learned by $M_{\mathrm{RLHF}}$ using GPT-4, which future work could leverage for reward modeling benchmarks or for training LLMs that learn more accurate reward models.

However, this method has several limitations as well. In LLMs larger than those used in these experiments (the largest of which was Pythia-410m), it may be required to explain many hundreds or thousands of features in order to effectively study their reward models. Both training autoencoders on activations of this scale and having GPT-4 explain the reconstructed activations becomes very computationally intensive. Furthermore, although features related to reward modeling can be extracted, the relationships of those features in producing a reward model remain unclear. Future work could focus on establishing these relationships for a more formal and broad interpretation of learned reward models in LLMs.

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

## A    FUTURE WORK

While our work presents evidence of sparse coding making reward models more interpretable, rigorous validation is still needed to ensure the technique provides faithful, complete, and minimal explanations. Future work should formulate more robust quantitative criteria to validate that the identified features accurately reflect the reward modeling process. Meeting such validation criteria would provide greater confidence that the technique yields rigorous and precise interpretations. Additionally, testing the approach on more complex reward modeling tasks is needed to understand its limitations and refine it towards minimal, complete circuits that faithfully reflect model computations. Specifically, future work could consider attempting to completely map the internal structure of a learned reward model using a basic unit like features, or perhaps one composed of circuits.

## B    REPRODUCIBILITY STATEMENT

In an effort to facilitate the reproducibility of our work, we have taken several measures to provide comprehensive resources. The weights for all the PPO models trained during this research will be made available in an open-source format via the Hugging Face model hub. The algorithms developed for feature extraction can be found in the paper's appendix. The source code for all experiments is available at this anonymized repository.

## C    RLHF WITH PROXIMAL POLICY OPTIMIZATION

We investigate the inner workings of a fine-tuned model $M_{\text{RLHF}}$, and contrast them to that of the equivalent base model $M_{\text{base}}$, which has only undergone pretraining. During fine-tuning, the model is subject to RLHF using Proximal Policy Optimization (PPO). This is achieved by having an evaluator review the model's outputs for a specified task and rate them. These ratings serve as the reward function $\text{Reward}(\tau)$, where $\tau$ represents a trajectory, a sequence of state-action pairs $(s_1, a_1, \ldots, s_T, a_T)$ for which $s_T$ represents text context at time $t$ and $a_T$ the token generated at that point.

In PPO, the objective is to maximize the expected sum of rewards $J(\theta)$, which can be defined as:

$$J(\theta) = \mathbb{E}_{\tau \sim \pi_\theta} \left[ \text{Reward}(\tau) \right] \tag{4}$$

Where $\pi_\theta$ represents the policy parameterized by $\theta$. The PPO algorithm optimizes this objective by updating the policy $\pi_\theta$ to a new policy $\pi_{\theta'}$ in a way that restricts the change in $\pi$. This is achieved by optimizing the following clipped objective function:

$$L(\theta, \theta') = \mathbb{E}_{\tau \sim \pi_\theta} \left[ \min \left( \frac{\pi_{\theta'}(a|s)}{\pi_\theta(a|s)} A_\theta(s,a), \text{clip} \left( \frac{\pi_{\theta'}(a|s)}{\pi_\theta(a|s)}, 1 - \epsilon, 1 + \epsilon \right) A_\theta(s,a) \right) \right] \tag{5}$$

Where $A_\theta(s, a)$ is the advantage function, and $\epsilon$ is a hyperparameter controlling the extent to which the policy can change. By employing PPO within the RLHF framework, the model iteratively refines its policies, thereby enhancing its performance and adaptability across a range of tasks.

Figure 2 is a graphic representation of the RLHF pipeline used to train $M_{\text{RLHF}}$.

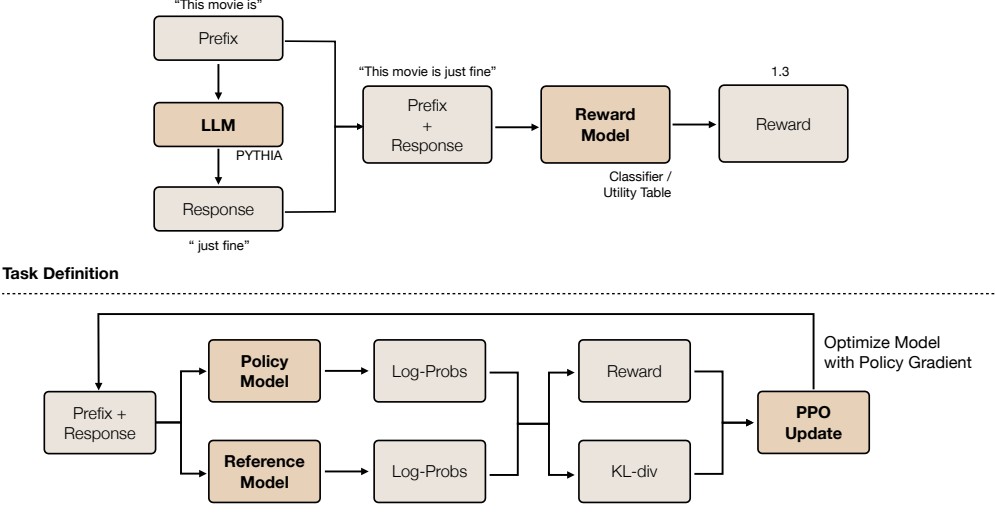

Figure 2: A prefix from the IMDb dataset is sampled as a prompt to a model from the Pythia suite, and then completed with the generation "just fine" in this case. Log probabilities are sampled from both the reference and policy model to compute the KL-divergence from the reference model, as well as compute the reward on the policy model's output distribution. This encompasses the complete training scheme for both the utility table and sentiment classifier tasks.

## D  OVERSEER-GUIDED FINE-TUNING USING UTILITY TABLES ARCHITECTURE DELINEATION

*Utility Designation*: Each word, represented as $w$, has an associated utility value defined as $U(w)$.

*Overseer* ($O$): A script that converts a tokenized sequence to words and takes a sum of their corresponding utility values in accordance with a utility table $U$.

*Student Model* ($M_{\mathrm{RLHF}}$): The model undergoing fine-tuning, shaped by feedback from the overseer.

*State* ($s$): Symbolizes a prompt or input directed to $M_{\mathrm{RLHF}}$.

*Action* ($a$): Denotes the response generated by $M_{\mathrm{RLHF}}$ corresponding to state $s$.

*Reward Mechanism*: For any generated action $a$, a sequence of tokens $t_1, t_2, ...t_n$, the reward Reward($a$) is calculated as Reward($a$) = $\sum_{i=1}^{n} U(w_i)$.

As is common in an RLHF setup, we train a policy model to maximize reward, while minimizing KL-divergence of generations from the reference base model otherwise. Here, $\pi_\theta(a|s)$ denotes the policy of $M_{\mathrm{RLHF}}$, which is parameterized by $\theta$, signifying the probability of generating action $a$ given state $s$.

## E   COMPLETE PYTHIA-70M FINE-TUNE TOP-K FEATURE DESCRIPTIONS

| Layer | Feature Index | Explanation |
| --- | --- | --- |
| 1 | 214 | looking for and activating upon the recognition of film titles or references to specific episodes or features within a series or movie. |
| 1 | 324 | looking for the initial parts of movie or book reviews or discussions, possibly activating on the mention of titles and initial opinions. |
| 1 | 433 | identifying and responding to language related to film and movie reviews or discussions. |
| 1 | 363 | looking for mentions of movies or TV series titles in a review or comment. |
| 1 | 208 | activating for titles of books, movies, or series. |
| 1 | 273 | looking for occurrences of partial or complete words that may be related to a person's name or title, particularly 'Steven Seag'al. |
| 1 | 428 | looking for unconventional, unexpected, or unusual elements in the text, possibly related to film or television content. |
| 1 | 85 | looking for negative sentiments or criticisms in the text. |
| 1 | 293 | detecting instances where the short document discusses or refers to a film or a movie. |
| 1 | 131 | 'The feature 131 of the autoencoder seems to be activating for hyphenated or broken-up words or sequences within the text data. |
| 2 | 99 | activating for hyphenated or broken-up words or sequences within the text data. |
| 2 | 39 | recognizing and activating for named entities, particularly proper names of people and titles in the text. |
| 2 | 506 | looking for expressions related to movie reviews or comments about movies. |
| 2 | 377 | looking for noun phrases or entities in the text as it seems to activate for proper nouns, abstract concepts, and possibly structured data. |
| 2 | 62 | looking for instances where names of people or characters, potentially those related to films or novels, are mentioned in the text. |
| 2 | 428 | looking for instances of movie or TV show titles and possibly related commentary or reviews. |

| Layer | Feature Index | Explanation |
|---|---|---|
| 2 | 433 | identifying the start of sentences or distinct phrases, as all the examples feature a non-zero activation at the beginning of the sentences. |
| 2 | 148 | identifying and activating for film-related content and reviews. |
| 2 | 406 | looking for broken or incomplete words in the text, often indicated by a space or special character appearing within the word. |
| 2 | 37 | activating on patterns related to names or titles. |
| 3 | 430 | detecting the traces of broken or disrupted words and phrases, possibly indicating a censoring mechanism or unreliable text data. |
| 3 | 218 | activating for movie references or discussion of films, as evident from the sentences related to movies and cinema. |
| 3 | 248 | identifying expressions of disgust, surprise or extreme reactions in the text, often starting with "U" followed by disconnected letters or sounds. |
| 3 | 87 | detecting the mentions of movies, films or related entertainment content within a text. |
| 3 | 454 | looking for general commentary or personal observations on various topics, particularly those relating to movies, locations, or personal attributes. |
| 3 | 46 | detecting strings of text that refer to literary works or sentiments associated with them. |
| 3 | 232 | identifying and focusing on parts of a document that discuss film direction or express a positive critique of a film. |
| 3 | 6 | looking for character or movie names in the text. |
| 3 | 257 | identifying the introduction of movies, actors, or related events.', 23: 'The feature at index 23 in an autoencoder appears to be looking for the beginning of sentences, statements, or points in a document. |
| 3 | 23 | looking for the beginning of sentences, statements, or points in a document. |
| 4 | 43 | looking for expressions of negative sentiment or criticism in the document. |
| 4 | 261 | looking for opinions or sentiments about movies in the text. |
| 4 | 25 | looking for the starting elements or introduction parts in the text, as all activations are seen around the beginning sentences of the documents. |
| 4 | 104 | activating on expressions of strong opinion or emotion towards movies or media content. |
| 4 | 38 | identifying statements of opinion or personal judgment about a movie or film. |
| 4 | 367 | identifying the expression of personal opinions or subjective statements about a certain topic, most likely related to movies or film reviews. |
| 4 | 263 | activating for statements or reviews about movies or film-related content. |
| 4 | 278 | activating for movie or TV show reviews or discussions, particularly in the genres of horror and science fiction. |
| 4 | 421 | identifying personal reactions or subjective statements about movies. |

| Layer | Feature Index | Explanation |
|---|---|---|
| 4 | 49 | detecting phrases or sequences related to storytelling, movies, or cinematic narratives. |
| 5 | 59 | looking for parts of text that have names or titles, possibly related to movies or literary works. |
| 5 | 76 | focusing on tokens representing unusual or malformed words or parts of words. |
| 5 | 156 | activating for the beginnings of reviews or discussions regarding various forms of media, such as movies, novels or TV episodes. |
| 5 | 236 | identifying critical or negative sentiment within the text, as evidenced by words and expressions associated with negative reviews or warnings. |
| 5 | 184 | detecting and emphasizing on named entities or proper nouns in the text like "Mexican", "Texas", "Michael Jackson", etc. |
| 5 | 477 | looking for reviews or comments discussing movies or series. |
| 5 | 284 | identifying the inclusion of opinions or reviews about a movie or an entity. |
| 5 | 454 | recognizing and activating for occurrences of names of films, plays, or shows in a text. |
| 5 | 225 | looking for phrases or sentences that indicate direction or attribution, especially related to film direction or character introduction in films. |
| 5 | 6 | identifying examples where historical moments, film viewings or individual accomplishments are discussed. |

## F    PSEUDOCODE

Algorithm 1 gives the pseudocode for both determining the most relevant layers, sparse autoencoder training, and finally automated feature interpretation.

## G    LAYER DIVERGENCES

Over here we graph the divergence of the RLHF-tuned models from the base model on a per layer basis, see Figure 3.

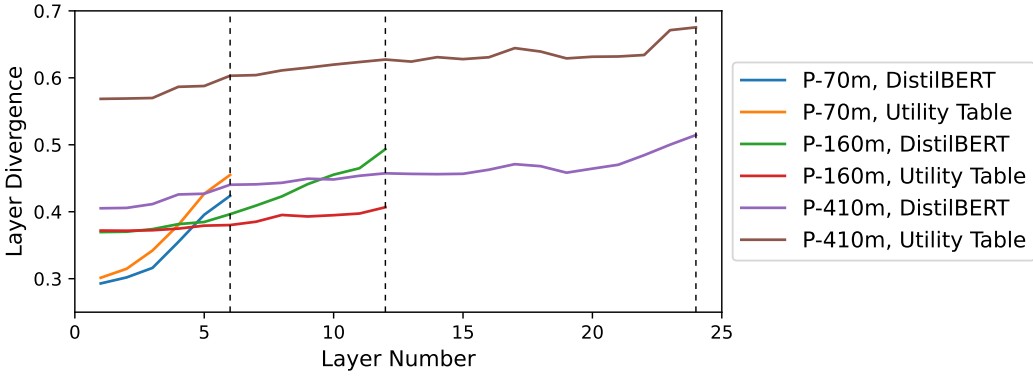

Figure 3: Divergences on a per-layer basis for various model and reward function combinations. Pythia-70m, Pythia-160m and Pythia-410m have 6, 12 and 24 layers respectively.

**Algorithm 1** Interpreting Learned Reward Models

1: **Initialize:** LayerDivergences ← []

**Step 1: Find top $n$ layers with most divergence**

2: **function** FIND_TOP_N_DIVERGENT_LAYERS(BaseLayers, FineTunedLayers)
3:      **for all** BaseLayer, FineTunedLayer ∈ zip(BaseLayers, FineTunedLayers) **do**
4:          Divergence ← COMPUTE $\Delta$(BaseLayer, FineTunedLayer)
5:          ▷ Divergence $\Delta$ is the sum of Euclidean distances between parameters.
6:          **Append** Layer and Divergence to LayerDivergences
7:      **end for**
8:      **Sort and Select** Top $n$ elements of LayerDivergences
9:      **return** LayerDivergences
10: **end function**

**Step 2: Sparse autoencoder feature extraction**

11: **function** EXTRACT_FEATURE_DICTIONARY(BaseModel, FineTunedModel)
12:      $D_{all}$ ← []
13:      **for** each Model in [BaseModel, FineTunedModel] **do**
14:          **for all** Layer in Top $n$ layers **do**
15:              $A$ ← GET_ACTIVATIONS(Model, Layer, TrainSplit)      ▷ Activation $A$
16:              $\mathcal{AE}_{\text{large}}$ ← TRAIN(Dictionary Size = 2*Activation Vector Dimension)
17:              $\mathcal{AE}_{\text{small}}$ ← TRAIN(Dictionary Size = Activation Vector Dimension)
18:              $D_{\text{large}}$ ← Decoder weights of $\mathcal{AE}_{\text{large}}$
19:              $D_{\text{small}}$ ← Decoder weights of $\mathcal{AE}_{\text{small}}$
20:              $D_g$ ← MMCS($D_{\text{large}}, D_{\text{small}}$)     ▷ MMCS finds high overlap dictionary $D_g$
21:              **Append** $D_g$ to $D_{all}$
22:          **end for**
23:      **end for**
24:      Return $D_{all}$
25: **end function**

**Step 3: Use feature weights to interpret inputs.**

26: **function** INTERPRET_FEATURE_INPUTS(BaseModel, FineTunedModel)
27:      **for** each Model in [BaseModel, FineTunedModel] **do**
28:          **for** each Layer in Top $n$ **do**
29:              **for** each top-$k$ Feature in Layer **do**
30:                  **for** each Review in IMDb Reviews Test Set **do**
31:                      Token_Subset ← First 50 tokens of Review
32:                      $A$ ← Activations of Model(Token_Subset)
33:                      $A'$ ← Activations of Autoencoder(A)
34:                      Top_Reviews ← Top 5 activating reviews for Feature
35:                      Explanation ← GPT-4(Top_Reviews, A')
36:                      DISPLAY(Explanation)
37:                  **end for**
38:              **end for**
39:          **end for**
40:      **end for**
41: **end function**

## H  METHODOLOGY FOR AUTOENCODER TRAINING

In this section, we discuss briefly various choices we made in the feature dictionary training setup as well as some light experimental exploration.

1. **The regularization $L_1$ coefficient ($L_1$-coef).** During autoencoder training, the sparsity of the feature dictionaries is enforced by adding an $L_1$ regularization loss on the feature weights, akin to Lasso (ZHANG & HUANG, 2008). We would ideally want $L_1$-coef to be low so as to allow the autoencoder training objective to reconstruct activation vectors with high fidelity using the dictionary features. But if $L_1$-coef is *too* small, then we observe an explosion in the "true" sparsity loss, namely the average number of non-zero positions in the dictionary features. These are then no longer as interpretable, and attend to almost all activation neurons.

   As such, we choose $L_1$-coef in a reasonable range to minimize both the true sparsity loss, as well as activation vector reconstruction loss. Empirically, we found a range of $L_1$-coef between 0.001 and 0.002 to be suitable in most cases. See Figure 4 for an illustration of the loss variation, over a single epoch of Pythia-70m trained with varying values of $L_1$-coef. We average the "true" sparsity loss over all highly divergent layers, and scale down by a factor of 100 for each in graphing.

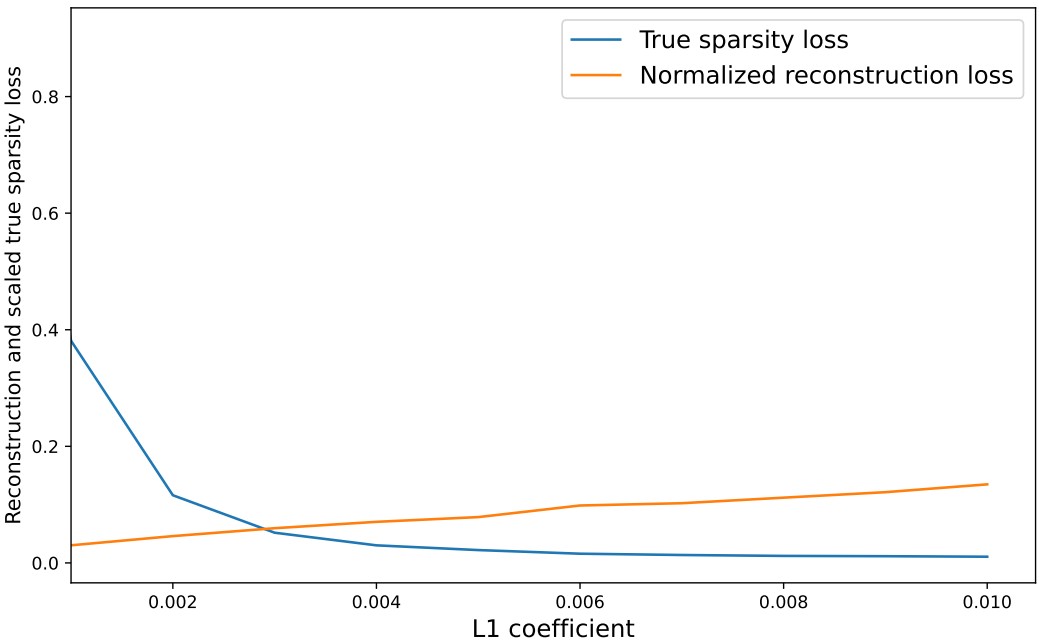

Figure 4: Normalized reconstruction and scaled true sparsity losses for Pythia-70m over 1 training epoch, over varying values of $L_1$-coef. Both metrics are averaged over all highly divergent layers, and hyperparameter choices are otherwise as described in Section 5.2.

2. **Tying encoder and decoder.** Another experiment choice we considered was whether to tie the encoder and decoder weights of the autoencoder. Tying the encoder and decoder weights has the advantage that each dictionary feature can then be explicitly written as a function of activation neurons. However, the model may be able to optimize the reconstruction and sparsity losses slightly better if the weights are left untied.

   We ran a small experiment on Pythia-160m and Pythia-70m with alternating the decoder and encoder weights as tied as well as untied. We found both the reconstruction loss and true sparsity loss to converge faster with tied weights. See Table 6. We suspect this trend may change with longer training times or different initialization schemes.

3. **How to select divergent layers.** In this paper, we have chosen to focus on the layers with highest parameter divergences. As can be seen in Section G and Figure 3, these tend to be the deepest layers of the neural networks. We briefly explored here the effects of looking at the lowest / initial layers of the neural networks instead.

| Model | Tied Weights | Sparsity Loss | Reconstruction Loss |
|-------|--------------|---------------|---------------------|
| pythia-160m | true | 0.291 | 0.053 |
| | false | 0.328 | 0.059 |
| pythia-70m | true | 0.383 | 0.030 |
| | false | 0.393 | 0.036 |

Table 6: Normalized reconstruction and scaled true sparsity losses for Pythia-70m and Pythia-160m over 1 training epoch, over differing choices of whether to tie encoder and decoder weights. Both metrics are averaged over all highly divergent layers, and hyperparameter choices are otherwise as described in Section 5.2.

Towards the end of our project, we ran a small experiment on Pythia-160m and Pythia-70m with alternating selecting the layers for autoencoder extraction as the lowest layers, vs the highest divergence layers. We found both the reconstruction loss and true sparsity loss to be far less for the lower most layers. A future study to examine the dictionary features extracted from these lowest layers would be interesting. See Table 7 for the observed metrics.

| Model | Divergence Choice | Sparsity Loss | Reconstruction Loss |
|-------|-------------------|---------------|---------------------|
| pythia-160m | highest ivergence | 0.291 | 0.053 |
| | lowest layers | 0.166 | 0.023 |
| pythia-70m | highest divergence | 0.388 | 0.036 |
| | lowest layers | 0.329 | 0.021 |

Table 7: Normalized reconstruction and scaled true sparsity losses for Pythia-70m and Pythia-160m over 1 training epoch, over differing choices of divergence. Both metrics are averaged over all highly divergent layers, and hyperparameter choices are otherwise as described in Section 5.2.

