# OpenReview forum: "Interpreting Reward Models in RLHF-Tuned Language Models Using Sparse Autoencoders"
_ICLR.cc/2024/Conference — ICLR 2024 Conference Withdrawn Submission_

### Official Review · Reviewer_UbRH · 2023-11-01

**Soundness:** 2 fair
**Presentation:** 2 fair
**Contribution:** 2 fair
**Rating:** 3
**Confidence:** 4

**Summary:**

This paper proposes a pipeline for finding and interpreting features in models trained with RLHF and the difference in learned features in the RLHF model vs the base model. They train sparse auto-encoders on model activations (as proposed by prior works) to decompose them into learned features, decomposing layers with the largest parameter difference, and then interpret the features learned by the models. They perform experiments on a movide review sentiment completion task. They provide some qualitative analysis of the learned features, and show that the features in the RLHFed model correspond more to the features the reward function was evaluating than the features in the base model, weakly implying that the RLHFed model has internalised the reward function to some extent.

**Strengths:**

* The topic of understanding the internals of RLHFed models is important, and the sparse auto-encoder approach hasn't been applied to this domain, so there is some novelty in that.
* The diagrams aid in understanding parts of the experimental setup.
* The measure of absolute utility of features is interesting and somewhat novel.

**Weaknesses:**

# Presentation and clarity

The paper's writing is somewhat difficult to follow at times, partially due to confusing use of terminology. The paper and title talks about interpreting reward models, but their approach interprets policy models, not reward models. If they are aiming to interpret a potentially implicitly learned reward function inside the policy model, that should be made more clear, and the assumptions underlying whether a policy model would internalise the reward function make explicitly.

# Unmotivated methodology

While the methodology is interesting, many design choices are made without any motivation, and it's unclear whether alternatives were considered and how well they performed. For example:
* choosing layers with higher parameter difference, rather than some other difference measure (e.g. in activations)
* training two autoencoders on each layer and taking features that occur only in both.
* The autoencoder architecture in general
* the choice of implementation of the Bills et al (2023) methodology.

While many of these choices occurred in previous work, it would be useful to justify and investigate their use in this work to ensure the methodology proposed is the best version it can be.

# Limited experiments and evaluation

Experiments are only performed in a single setting (movie review completion), with a range of relatively small models. Further, even within this setting the analysis is very limited, comprising just a qualitative analysis of some of the features present (table 3 and section 6.1), and the measure of aggregate absolute utility in table 4. Given this, it's unclear whether the proposed methodology achieves what it is intended to achieve (and it is somewhat unclear what this goal is), and what we have learned about implicit reward functions in RLHFed models from this work.

I suggest the authors perform more extensive experiments, more evaluation of their methodology vs possibly ablations and alternatives, and try to provide more concrete conclusions that the analysis shows about RLHFed models.

# Summary

Overall, I don't feel this work is ready for publication at at top venue such as ICLR. I'd encourage the authors to spend more time improving the clarity, presentation and motivation of the work, as well as performing additional analysis and experimentation to validate their approach and gain clearer conclusions from the analysis. This is an important topic, and I believe an much-improved version of this paper would be impactful and important to the community.

**Questions:**

* This is mostly addressed previously, but for each of the design decisions made (i.e. the bullet points in section 4.1), describing the motivation in more detail for choosing this approach vs other possible alternatives would be beneficial.
* What's the strongest or most exciting conclusion from the analysis in this work?

---

> ### Author Response · Authors · 2023-11-21
> **Response to Reviewer 3.**
>
> We thank the reviewer for their thorough, and careful responses to our paper, and outline the changes we have made to address these concerns.
>
> ### Regarding presentation and clarity
> We are indeed focusing on the implicit reward models learned by our policy models, as the reviewer suggests. We now clarify this right from our abstract and do a thorough pass over the paper standardizing and clarifying where we are referring to:
> * The desired reward model which is the objective of the RLHF training, or
> * The implicit reward model which is learned by the policy model.
>
> We have made an extensive pass clarifying language, notation and presentation throughout, and updating all of our figures.
>
> ### Regarding unmotivated methodology
> In response to the reviewer’s comment about better motivating our design choices, since our initial submission to ICLR and further spurred by the reviewer’s kind suggestion, we have made the following additions:
>
> * We explore the efficacy of these different choices for our autoencoder setup via two different metrics namely:
>     * The reconstruction error in the activation vector, and sparsity of the features that are extracted.
>     * The average utility of the top-k feature descriptions, that we presented in the original draft of our paper.
>
> Using the above two metrics as our objectives, we have added the following explorations to our paper:
> * We add experiments for selecting layers to extract autoencoder features based on different choices, including:
>     * Highest parameter divergences, as in the original draft of the paper.
>     * Lower most layers, as suggested in the cited work of “Sparse autoencoders find highly interpretable features in large language models”, by Cunningham et. al.
>
> We utilize these same metrics above to explore a sampling of choices in the autoencoder setup, namely:
> * The choice of whether weights should be tied between the encoder and decoder of the autoencoder.
> * The L1 “sparsity” loss on the feature dictionaries.
>
> Regrettably, we are not able to engage in a deep study of the Bills et al methodology for auto interp, primarily due to the cost and latency involved with running high volumes of GPT-4 queries.
>
>
> ### Regarding comprehensiveness of models and tasks
> * We have added results for feature dictionaries on a GPT-NEO model. In addition, we commit that we will include autoencoders for GPT-J 7B in a camera ready version.
>
> * Along with the results for Pythia in the original submitted draft of the paper, this would give us results for 3 different model architectures of varying sizes
>
> * While we readily concede this is still not fully exhaustive, we note that for instance the original paper by Cunningham only studied one model size the Pythia architecture, and that we have been able to validate the general method works in a broader context.
>
> * To show the utility of our method is not confined to sentiment based text generation, a camera ready version would include an appendix section with a small case study for the very widely used “Helpless and Harmless” hh-rlhf dataset from Anthropic.
>
> For transparency, we clarify that the GPT-J and hh-rlhf case study would be done before the camera ready version, but are not likely to make the 22nd revised submission deadline.
>
> ### Regarding limited experiments and evaluation
> We extend our quantitative analysis by:
> * Demonstrating correlation of the summed reward utility of top-k extracted features, with the original utility scores of generated text.
> * Using the top-k extracted features to find a reward modeling failure.
>
> In general, these added results demonstrate more suggestively than the original draft, that the extracted features can be used both as a rough proxy to the quality of the implicitly learned reward model, as well as a means to expose reward modeling features.
>
>
> ### On questions posed by Reviewer
> * As mentioned earlier in this review, we flesh out at least a sampling of the methodology choices outlined in Section 4, including different ways to select layers and permutations on the autoencoder setup.
> * On what we believe to be the most exciting result,  we believe it is the (now further examined) correlation between the performance of the trained policy model on the RLHF objective, with the measured utility of the sparse coding feature descriptions extracted via GPT-4 on the activations of the sparse codes.

---

> > ### Comment · Reviewer_UbRH · 2023-11-22
> > **Upload new revision**
> >
> > I thank the authors for their thorough response. However, the current revision on openreview has not been updated, so it is difficult to assess the changes you have made. Could you upload the new revision so that I can assess that?

---

> > > ### Author Response · Authors · 2023-11-22
> > > **Working draft uploaded**
> > >
> > > We've uploaded a working draft now, which includes all of the writing, formatting and presentation fixes.
> > >
> > > We are still writing up a couple of the experiment results mentioned in the responses - I apologize for having gotten confused slightly on the timelines expected - but expect to have a more up to date draft uploaded later today.

---

### Official Review · Reviewer_WBtF · 2023-11-03

**Soundness:** 2 fair
**Presentation:** 1 poor
**Contribution:** 3 good
**Rating:** 3
**Confidence:** 5

**Summary:**

The paper studies the interpretation of reward models in RLHF. The authors take high divergence layers, and train sparse autoencoders to find high confidence features, and analyze the activations on these features.

**Strengths:**

I like the idea of using autoencoders to help find important features, and analyze activations on these features via asking GPT-4 to give explanation. Although similar methods have been applied to GPT-2, there haven't been methods explaining reward model. It helps us better understand how the reward model

**Weaknesses:**

The writing and experiment design of the paper can be significantly improved. Besides the formatting issue which shall make the paper be desk rejected, the English and mathematical notations are very confusing and not well defined. For example, in the beginning of page 6 the authors are using tokens and words interchangeably, with both $t_n$ and $w_n$. There are also large amount of typos in the paper.

The authors mention that the pearson correlation serves as a grading for the accuracy of the prediction. But there is no such experimental results reported.

The paper can also be significantly improved if the authors could select some of the GPT-4 explanations, and design some specific prompt-response pairs to validate the GPT-4 descriptions.

Overall, I find the idea interesting. But the current writing and results are definitely not enough for ICLR standard. Thus I recommend rejecting the paper for now and wait for a better version in the future.

**Questions:**

See weakness.

---

> ### Author Response · Authors · 2023-11-21
> **Response to reviewer 2.**
>
> We thank the reviewer for their thorough and careful responses to our paper, and outline the changes we have made to address these concerns.
>
> ### Correcting notation
> As suggested, English notation has been unified throughout the paper (for example in the case of the token/word confusion, as well as the reference to reward models, for which we provided an acronym for the phrase “implicit reward model” (the object of our study)).
>
> ### Removing typos
> We corrected a large number of typos where they were identified, as well as some formatting issues with the original paper that made the mathematical notation confusing. (See the response to reviewer 1 for more details on the typos removed.)
>
> ### Adding more results
> As suggested by the reviewer, results have been updated to include the Pearson Correlation Coefficient in the appendix, and some additional experiments  have been provided to support the conclusion of the paper:
>     * We search for specific features changes caused by fine-tuning that contradict the provided table of utility values for a concrete example of a reward modeling failure.
>     * We correlate the utility of the description of the top-k features with the average utility of generations over a corpus of text, to show the relationship between an increase in the utility of the description of the top-k features and the model’s propensity to generate high utility completions.
>
> ### Validating GPT-4 descriptions
> The process of computing the Pearson Correlation Coefficient are now included in the appendix of the paper, and hopefully constitute validating GPT-4 descriptions. As explained in the updated paper version and in prior work, the Pearson Correlation Coefficient, involves simulating activations if the provided description were true, and then computing the difference between the simulated and true activations over a corpus of text. As such, this is an explicit quantitative measure of the validity of GPT-4 feature descriptions.
>
> Once again, we would like to thank the reviewer for their careful feedback and helpful suggestions.

---

### Official Review · Reviewer_Btqy · 2023-11-08

**Soundness:** 3 good
**Presentation:** 2 fair
**Contribution:** 3 good
**Rating:** 3
**Confidence:** 3

**Summary:**

In this paper, the authors aim to provide some insight into what a RLHF-finetuned model learns during the RLHF procedure, with particular attention placed on the *implicit* reward function learned while performing RLHF (against an explicit reward model).

To do so, they employ a multi-step procedure, whereby a model is first tuned via RLHF. Then, layers which have changed significantly due to RLHF (in Euclidean norm) are identified, and autencoders (of two different sizes – one pair per layer) are trained to reproduce activations in these layers in both the RLHF'd model and the original model. Relevant features are extracted by seeing which ones occur in the larger and smaller autoencoders. Finally, via a procedure I have yet to fully understand, feature descriptions are provided by GPT-4 to explain which the features are actually doing.

This paper represents a first step towards interpreting the *implicit* reward model learned by a LLM during RLHF (NOT the reward model used for RLHF), which is a very important direction of study given the dominance of RLHF-based finetuning in modern-day ML.

**Strengths:**

## significance

Although I will be quite critical in this review in general, I want to start out by stressing how happy I am to see people working on this topic. In general, the field is in dire need of more people who are aware of and working on AI safety related issues. Regarding the area of study of this paper in particular – understanding reward functions implicitly learned during RLHF – this is, in my mind, a super important and relevant area, and continued research in this area could have a meaningful impact in improving the chances that AGI goes well. In this regard, I applaud the authors and want to again express my excitement with and belief in the general direction.

In particular, the opening question about whether reward models (perhaps I would prefer: "objectives" or "goals") learned during RLHF actually correspond to the training objectives very compelling.

## originality

The method suggested in the paper appears to be fairly original. I am not aware of work exploring the implicit reward models learned during RLHF (assuming they exist).

## quality

There was clearly effort put into this research, and I with more work (especially on the paper presentation) I think it can be of meaningful quality.

## clarity

The background and related work sections are quite clear and pleasantly extensive. If such clarity is brought to the rest of the paper – especially sections 4, 5, 6 – then it will be very good.

**Weaknesses:**

## Margins

The margins appear significantly smaller than ICLR requirements.

## Falling short of answering the key question

The key question being asked in the intro, which I find very compelling, is about whether RLHF actually teaches the model the correct thing, and whether we can interpret how they fall short of this. Yet, the paper is not quite about this topic – it's more about interpretability of learned reward in general, without any comparison to the original reward model or divergences with it. That's ok, but it should be clear that you're trying to answer on a subset of the question, since you provide the question as the main motivation.

## Paper layout

I found the different sections a bit hard to follow – it was unclear what was going to discussed next. For example, the distinctions between sections 4, 5, and 6 were not so clear to me.

## General confusion

I found myself quite confused at several parts in the paper, and I think part of this is due to the presentation. One major point of confusion for me is that there seems to be a conflation (or in any case, and ambiguous distinction) between the (explicit) reward model used during RLHF and the (implicit) reward model which is (maybe) learned as a result of RLHF. In particular, most of the discussion, and all the experiments, are focused on understanding the implicit reward model. Yet, to my knowledge, it is not even guaranteed that there _exists_ such an implicit reward model – that could be one way that the RLHF'd agent learns to get higher rewards, but it could also simply learn a value function and/or policy without any implicit computation of reward itself.

Personally, I would encourage the use of "implicit reward model" always, vs "learned reward model", since the actual reward model learned from human preferences is also learned during RLHF, which causes confusion.

## Reproducibility

It would be good to have access to the code during the review procedure so reviewers can reproduce the results / examine the experiment setup. Since anonymity is a concern, an anonymizing tool can be used, such as: https://anonymous.4open.science/.

## Algorithm presentation

It would be good to make the comments (like "Find top n layers with most divergence) different colours for clearer presentation

in line 9, what is the input that you're calculating the activations over?

in lines 10 and 11, shouldn't the autoencoder size be smaller than the A layer dimension? Otherwise, what compression is happening?

12 and 13: I propose using the convention A\Epsilon_\large.Decoder. Otherwise it looks like you're applying a function called Decoder, whereas here I believe you're trying to extract the decoder from the autoencoder? Please tell me if I'm misunderstanding here.

21: 50 tokens seems quite long. In the paper you gave an example with 3 tokens. Was 50 what you actually did in the experiments? Having access to the codebase would help answer this kind of question.

consider using \texttt for variable names, since that leads to nicer treatment of underscores

## Typos/unclear bits

There are a medium number of typos/unclear bits present in the paper:
- approximation reward model -> approximation of reward model
- second paragraph "Elhage et al" should be \citep
- model activations were sampled -> model activations that were sampled from
- a L_1 coefficient -> an L_1 coefficient
- proposed specifically for expressing reward models (I don't know what "expressing a reward model" means)
- compared to features out brought -> compared to features brought
- in section 4.2, instead of saying "below", the paper should reference a Table by its number
- the table "below" should have a number and a caption. It could go in a wrapfigure if space is a concern.
- the formatting of the list of architecture components in section 4.2 is broken
- gnerations -> generations
- in 5.1, the Sanh et al should be \citep
- it would be nice to find dates for the two von Werra references
- Table 1: Hyperparameters used in the experiment. Which experiment?
- Figure 2: model model -> model
- 5.2 you mention "the previous experiment". Which one is that?
- an autoencoder per high-divergence -> one autoencoder for each high-divergence
- "the following dictionary sizes" (again, reference the Table specifically)
- learned these divergences below (reference the Table specifically). also, please mention that the table is only a subset of the whole table, and ideally include things not only from layer 2.
- e.g. `good', `happy', etc -> e.g., `good', `happy', etc.
- GPT-4s -> GPT-4's
- lend well to this task -> lend themselves well to this task
- utlity -> utility

**Questions:**

## General confusion

- "Our primary method for interpreting reward learned reward models consists of first isolating parameters relevant to reward modeling" (I'm not sure what this sentence means. Are you talking about explicit or implicit reward model? And why are parameters being discussed here – I thought the analysis was done at the level of features, not parameters?

- "such that any changes in learned features due to RLHF are interpretable" (what does this mean?)

- "were used between all dictionary pairs" (what are all dictionary pairs?)

- when you say "from high likelihood layers involved in reward modeling" do you mean "from layers which are highly likely to be involved in reward modeling"?

- "through the spares coding method from the previous exporiment" (which previous experiment are you referring to?)

- when you train the autoencoders, where are you getting the activations from? a random sample of the dataset? the whole train set?

- how are you getting the feature descriptions from GPT-4? I follow the procedure up to step 4 (on page 2), but I'm not sure where the GPT-4 descriptions come from.

- you mention "selecting case studies of reward modeling failures and successes to show the utility in doing so" but I'm not sure where you show these case studies.

- since the RLHF objective was to give *positive* reviews, I would have expected the interpretability method to find features related to whether a review was good or bad. Yet, it seems most features have to do with movie stuff in general, which is to be expected simply by the nature of the dataset. Based on this experiment, to what extent can you say that the method is useful for finding features learned by RLHF, and not just finding somewhat random features pertaining to the dataset which were learned during fine-tuning? Perhaps section 6.2 aims to explain this, but I do not understand what it is saying.

## Statements in the paper

- You talk about finding the high probability layers involved in reward modelling by looking at the parameter divergence between base model and fine-tuned model. Yet, how do you know which parts of that divergence are used in reward modelling (if any), and which are used in other things, such as learning a policy, or planning, or building a world model, etc?

## Experiment suggestions

- I'm curious why you chose to use exactly two autoencoders. It would be nice to have an experiment with three (or more) and to show how much more information is gained by doing so.

## Closing comment

I want to again emphasize the importance of this topic and my encouragement to the authors to please continue pursuing this and related lines of research. Thank you for working on AI safety related topics. I hope that my and others' comments can help this paper become an impactful part of the AI safety literature.

I would happily consider changing my rating based on the responses of the authors to my questions, reformatting of the paper to match ICLR guidelines, the provision of experiment code, and increased clarity in sections 4, 5, and 6. As it stands, as it is written at present, I would give the paper a 4 (not a 3) but that doesn't appear to be an option I can select.

---

> ### Author Response · Authors · 2023-11-21
> **Response to reviewer  Btqy - [1/2]**
>
> We thank the reviewer for their thorough and careful responses to our paper, and outline the changes we have made to address these concerns:
>
> ### Clarity
> * Sections 4, 5 and 6 have been rewritten with clarity in mind, and so that each section is more tightly focused. We now open each section/subsection with why its contents are relevant and finish those sections with a one sentence recap of what the takeaways are from that section.
> * Section 4 now gives an overview of the methodology and approach. Section 5 gives all the experiment pipeline details, namely the external reward model definitions, the model hyperparameters for both the RLHF stage as well as for the autoencoder training. Section 6 gives the qualitative analysis of case studies of feature dictionaries, and the quantitative results measuring the summed utility of these via GPT-4 .
> * We also re-organize material from sections 5 and 6, so that section 5 focuses on the experiment setup and parameters, and section 6 focuses on the case studies and results, whereas before they were less distinct.
>
> ### Margins
> The issues with margin size have been corrected.
>
> ### Paper layout
> As mentioned prior, we hope to have corrected this via the new structure we adopted for these sections, as well as the reorg of sections 5 and 6.
>
> ### General confusion
> By adopting the suggested naming scheme of ‘implicit reward model’ (which we abbreviate to IRM), we hope to have addressed the ambiguity between the reward models involved. For example:
> “Do implicit reward models (IRMs) learned by Large Language Models (LLMs) through Reinforcement Learning from Human Feedback (RLHF) diverge from their intended training objectives?”
> “As LLMs steered via RLHF scale in capability and deployment, the implications of failures in the IRM amplify.” In general we clarify this paper focuses on implicit reward models, and adjust terminology usage accordingly.
>
> As for the concern with the use of the term ‘reward model’ when the model; we use the term not to refer to implicit internal computation of reward, but rather the presence of explicit linear structure within the activations that correspond to the fine-tuning objective. We have added an experiment that shows correlations between these and the actual reward.
>
> ### Falling short of answering the key question.
>
> We have adjusted the abstract to account for our focus on interpreting the implicit reward model, rather than on correspondence with the desired reward model.
> We also include additional results that explicitly do illustrate reward model correspondence by correlating the utility of model generations with the utility of the GPT-4 descriptions of the top-k features.
> We still soften the claim in the abstract, as we can’t yet fully answer this reward model correspondence question, but hope this makes our answer to said question more substantial.
>
> ### Reproducibility
> Source code has since been provided anonymously at https://anonymous.4open.science/r/rlhf-C7C8/. We will continue to update and clean this repository before any camera ready version.
>
> ### Algorithm presentation
> * We adopted the reviewer’s suggestion of additional colors for natural language details in the algorithm in the appendix, and used the suggested convention to convey extracting the dictionary from the autoencoder. We also specify that the activations for the autoencoder were from the train split of the imdb dataset.
> * “in lines 10 and 11, shouldn't the autoencoder size be smaller than the Ativation layer dimension?”  The power of the sparse autoencoder is not so much in dimensionality reduction, but rather the sparse linear structure of the autoencoder disentangling superposition in the original MLP. In fact, Anthropic’s blog on sparse coding published after our abstract submission scaled up the hidden size dimension by 256x! (See https://transformer-circuits.pub/2023/monosemantic-features/index.html).
> * We originally had language in the paper suggesting the power was in the dimensionality reduction. We apologize for this source of confusion, and have since clarified the verbiage.
>
> We respond to the remainder of the review in our next comment.

---

> ### Author Response · Authors · 2023-11-21
> **Response to reviewer Btqy  typos and general questions Btqy - [1/2]**
>
> We continue our response here to Reviewer 1's very thoughtful and helpful feedback.
>
> ### On typos and unclear bits:
>
> All identified typos (and others found after multiple re-reads) have been corrected in accordance with the reviewers suggestions.
> In addition we have made all references to tables and figures explicit, rather than positional.
>
>
> ### On General Confusion:
> By adopting the suggested naming scheme of ‘implicit reward model’ (which we abbreviate to IRM), we hope to have addressed the ambiguity between which reward models are being referred. For example:
> “Do implicit reward models (IRMs) learned by Large Language Models (LLMs) through Reinforcement Learning from Human Feedback (RLHF) diverge from their intended training objectives?”
> “As LLMs steered via RLHF scale in capability and deployment, the implications of failures in the IRM amplify.”
>
> As for Reviewer 1's concern with the use of the term ‘reward model’, when the model may be learning some sort of "value function". We clarify that we use the term here not to refer to implicit internal computation of reward, but rather the presence of explicit linear structure within the activations that correspond to the fine-tuning objective.
>
> ### On statements in the paper.
> Parameter divergence was used to discard layers with little divergence to avoid running unnecessary experiments. Training autoencoders on activations from layers with such little change from the base model, that their probability of containing components of IRM was small. This is admittedly a choice also based in convenience and practicality.
>
> ## On experiment suggestions
> We chose two autoencoders, because the cosine similarity metric that has garnered popularity in determining the most effective dictionary features is a pair wise measure, only applicable when two autoencoders are trained.  (We hope this is more clear in our rewritten submission.)
>
> This measure could be modified/extended, but it would be a significant and compute intensive perturbation to existing methods, which we were gauging the effectiveness of for reward model interpretation. We agree that it would be interesting to experiment with more autoencoders and alternative similarity metrics in the future.
>
>
> In general, we thank the reviewer again for their helpful insights, and hope our response demonstrates we have taken the suggestions seriously

---

> ### Comment · Reviewer_Btqy · 2023-11-23
> **Thank you for your responses**
>
> ### Response
>
> I thank the authors for their responses to my comments and questions. The reformatting and fixing of typos is great (I only see two typos now: "Seeing appendix", and "Elhage et al" should be \citep).
>
> I'd also like to compliment the authors on their general code style and the presence of docstrings. I'd gently encourage more use of dataclasses, which can reduce clutter. I'd also like to express my appreciation for section 6.1, which recognizes important shortcomings of the approach as is. Finally, I also quite like the new visuals, which add to the clarity of the presentation.
>
> All this said, I still feel like there is more work to be done before this paper is quite at the level I feel comfortable recommending (to give a numerical response, I'd say it's now a 4.5). This is primarily due to an ongoing sense of general confusion (to be clear, it is better than it was before). An illustrative example: a utility table U is mentioned in section 4.2, but as far as I can tell doesn't appear in the paper or appendix. Then in section 5.1, the authors mention giving higher rewards to positive sentiment prefix-completion pairs – what happened to the utility table task?
>
> ### Misc questions/thoughts
>
> - which checkpoints were used for the Pythia models?
> - is it obvious that the Bills et al. approach is well-suited for this setting (models of this size)?
> - You say that “good” and “happy” should be more prevalent in the feature *descriptions*, but it seems like what you should care about is whether the feature descriptions measure positive/negative sentiment? Like the description could be “this sentence contains 'good/bad'” or “this feature senses positivity/negativity in the review” and that would be relevant. But if the description says "this feature recognizes reviews which have good writing" then that would contain the word "good" but not be the kind of feature you're looking for.
> - What’s the absolute utility of the top-k most similar feature descriptions, and how do you get it?
> - How do you measure similarity between feature descriptions?
> - from above: a utility table U is mentioned in section 4.2, but as far as I can tell doesn't appear in the paper or appendix. Then in section 5.1, the authors mention giving higher rewards to positive sentiment prefix-completion pairs – what happened to the utility table task?
> - the IRM/ERM distinction is much clearer than before, but there are still some places where "reward model" is used to (I think) mean IRM.
> - it would be good to have more justification of why we can assume that IRMs are definitely a thing that's being learned during RLHF.
>
> ### Conclusion
>
> I hope the authors can find this encouraging vs discouraging, as in just a week, it seems significant progress has been made. I wish the authors the best of luck with this research!

---

> > ### Author Response · Authors · 2023-11-23
> > **Thank you!**
> >
> > Since the period to leave responses/comments is almost at an end, we just want to take a moment to thank the reviewer for all their helpful insights and suggestions!
> >
> > We understand it isn't always appealing to deal with typos or presentation mistakes, and really appreciate the effort you've put in. It has been helpful for us in making several cleanups, reorganizations and improvements.